# Chemically Crosslinked Sulfonated Polyphenylsulfone (CSPPSU) Membranes for PEM Fuel Cells

**DOI:** 10.3390/membranes10020031

**Published:** 2020-02-18

**Authors:** Je-Deok Kim, Akihiro Ohira, Hidenobu Nakao

**Affiliations:** 1Hydrogen Production Materials Group, Center for Green Research on Energy and Environmental Materials, National Institute for Materials Science (NIMS), 1-1 Namiki, Tsukuba, Ibaraki 305-0044, Japan; 2Energy Storage Technology Group, Research Institute for Energy Conservation, National Institute of Advanced Industrial Science and Technology (AIST), 1-1-1 Higashi, Tsukuba, Ibaraki 305-8565, Japan; a-oohira@aist.go.jp; 3Hydrogen Materials Engineering Group, Research Institute for Energy Conservation, National Institute of Advanced Industrial Science and Technology (AIST), 1-1-1 Higashi, Tsukuba, Ibaraki 305-8565, Japan; nakao.hidenobu@nims.go.jp

**Keywords:** PPSU, High IEC, CSPPSU, activation, PEMFCs

## Abstract

Sulfonated polyphenylsulfone (SPPSU) with a high ion exchange capacity (IEC) was synthesized using commercially available polyphenylsulfone (PPSU), and a large-area (16 × 18 cm^2^) crosslinked sulfonated polyphenylsulfone (CSPPSU) membrane was prepared. In addition, we developed an activation process in which the membrane was treated with alkaline and acidic solutions to remove sulfur dioxide (SO_2_), which forms as a byproduct during heat treatment. CSPPSU membranes obtained using this activation method had high thermal, mechanical and chemical stabilities. In I-V_iR free_ studies for fuel cell evaluation, high performances similar to those using Nafion were obtained. In addition, from the hydrogen (H_2_) gas crossover characteristics, the durability is much better than that of a Nafion212 membrane. In the studies evaluating the long-term stabilities by using a constant current method, a stability of 4000 h was obtained for the first time. These results indicate that the CSPPSU membrane obtained by using our activation method is promising as a polymer electrolyte membrane.

## 1. Introduction

A low-carbon society in which safe, highly efficient, renewable and sustainable energy sources are used to sustain economic growth, environmental protection and energy security has become important. Proton exchange membrane fuel cells (PEMFCs) are among the most promising electrochemical devices for a low-carbon society and highly efficient power generation. Although PEMFCs using perfluorosulfonic acid polymer membranes have been commercialized, to increase their performances, more development and new material components, such as catalyst electrodes and polymer membranes, are required. Although it is important to find non-platinum catalyst materials for use as the catalyst electrode [1,2,3], non-fluorine proton exchange membranes are also necessary. The most commonly used proton exchange membranes for PEMFCs are poly(perfluorosulfonic acid) (PFSA) copolymers such as Nafion, which have high hydrolytic and oxidative stability and excellent proton conductivities [4,5]. However, the glass-transition temperatures (T_g_), mechanical stabilities, and gas permeabilities must be improved. Moreover, perfluorinated polymers have high production costs and environmental incompatibilities. The drawbacks of perfluorinated membranes have prompted research into alternative membranes. For example, several aromatic polymer ionomer membranes, such as polybenzimidazole (PBI) [6,7], sulfonated polyphenylene oxide (SPPO) [8], sulfonated polyimide [9,10,11], sulfonated polyethersulfone (SPES) [12,13], sulfonated polyphenylene (SPP) [14], sulfonated polyphenylene sulfone (SPPS) [15], sulfonated polysulfone (SPSU) [16,17,18,19,20,21], sulfonated polyetheretherketone (SPEEK) [22,23,24,25,26,27,28,29,30,31,32,33], and sulfonated polyphenylsulfone (SPPSU) [34,35,36,37,38,39,40,41,42,43,44,45,46,47,48,49], are being actively investigated. The rigid molecular chain structures of aromatic polymers provide high thermal stabilities. However, their conductivities are generally lower than those of PFSA also because the structures can make it difficult to form proton conducting channels. However, increasing the ion exchange capacity (IEC) of the aromatic polymer to increase the conductivity results in weaker mechanical and chemical properties of the polymer. In other words, there is a trade-off relationship between the increase in conductivity and the mechanical and chemical properties. Thus, further development of hydrocarbon polymers without fluorinated moieties is necessary.

Polyphenylsulfone (PPSU), which has an excellent thermal stability, high chemical resistance and low cost, has been studied widely. The sulfonation of PPSU to balance the IEC values and membrane stability is a crucial focus. High IEC values are desirable for good proton conductivities but result in excess swelling, which causes mechanical and chemical instabilities. These mechanical and chemical properties are big obstacles for applying SPPSUs in PEMFCs. We are developing a crosslinked sulfonated polyphenylsulfone (CSPPSU) membrane having good thermal and chemical properties with good proton conductivities [34,36,37]. Organic solvent-free CSPPSU membranes have been reported [34,37]. Moreover, an SPPSU with a high sulfonation degree from bis(4-fluorophenyl)sulfone (FPS) monomer has been used to prepare a CSPPSU membrane [36]. The membranes have high proton conductivities (>0.1 S/cm at 90% RH, 80 °C) [34,36,37]. However, the cell performances are not stable over a long period. We thought that a byproduct such as sulfur dioxide (SO_2_) produced during the crosslinking process remaining in the membrane was the cause. In this paper, we report an activation treatment for the CSPPSU membrane to achieve good thermal, chemical, and long-term stabilities. The nanostructures, conductivities, cell performances, and thermal, mechanical and long-term stabilities of the CSPPSU membranes after our activation treatment were investigated for the first time and compared with those of Nafion212 membranes.

## 2. Experimental

### 2.1. Materials

PPSU (Solvay Radel R-5000 NT) (Mn = 26,000; Mw = 50,000; Mw/Mn = 1.9) was provided by Solvay Specialty Polymers Japan K.K. (glass transition temperature (T_g_) = 220 °C). A DuPont^TM^ Nafion^®^ PFSA membrane (NR-212) was purchased from DuPont (USA). The chemical compounds were purchased from commercially available sources and were used as received. Hydrogen peroxide (H_2_O_2_), sodium chloride (NaCl), sodium hydroxide (NaOH), and sulfuric acid (H_2_SO_4_) were purchased from Nacalai Tesque, Inc, Japan. Iron (II) chloride tetrahydrate (FeCl_2_·4H_2_O) was purchased from Wako Pure Chemical Industries, Ltd., Japan. A dialysis tubing cellulose membrane (molecular weight cut-off; MWCO = 14,000) and dimethyl sulfoxide (DMSO) were purchased from Sigma-Aldrich Co., Ltd. Deionized (DI) H_2_O was obtained using a PURELAB^®^ Option-R 7 ELGA LabWater at 15 Mohm cm and 25 °C.

### 2.2. Synthesis and Characterization of SPPSU

SPPSU was synthesized using an electrophilic aromatic substitution reaction of PPSU with H_2_SO_4_. The synthesis and properties of SPPSU have been described in detail in a previous paper [34]. The SPPSU polymers had IEC values of 3.68 meq/g, and the degree of sulfonation (D.S.) values were 2.3, which are close to the theoretical IEC value of 3.57 for D.S. = 2.0. The yield was 76%.

### 2.3. Preparation of Chemically Crosslinked SPPSU (CSPPSU) Membranes

SPPSU (5 g) was dissolved in DMSO 20 mL with stirring at 60 °C to prepare a 20 wt% solution. The solution was casted at a speed of 4.0 mm/min with an applicator (blade) (Tester Sangyo CO., Ltd., Japan; 15 cm, gap = 0.5 mm) on a glass plate (27 × 30 cm^2^) heated at 80 °C by an automatic film-coating apparatus (KIPAE, KP-3000VH), and then dried for 24 h at 80 °C. Next, the film coated on the glass plate was moved to an oven and annealed in air at 120 °C (24 h), 160 °C (24 h), and 180 °C (24 h). After that, the membrane was peeled off the glass plate with water and activated to remove the remaining sulfur dioxide (SO_2_), which is a byproduct from the annealing process, and to cure the nanostructure of the CSPPSU membrane. Activation was performed using the following procedure: heating in 0.5 M NaOH at 80 °C overnight, boiling in DI H_2_O for 2 h, heating at 1M H_2_SO_4_ at 80 °C for 2 h, and boiling in DI H_2_O for 2 h. Finally, the CSPPSU membranes were dried at room temperature before using. The CSPPSU membranes were very flexible and light brown.

### 2.4. Pretreatment of Nafion212 Membranes

Nafion212 membranes were prepared for a comparison with the CSPPSU membranes. Before using, the Nafion212 membranes were pretreated as follows: they were boiled in DI H_2_O for 2 h, heated in 1M H_2_O_2_ at 80 °C for 2 h, followed by heating in 1M H_2_SO_4_ at 80 °C for 2 h, and then boiled in DI H_2_O for 2 h. The membranes were kept in DI H_2_O until use.

### 2.5. IEC, D.S., Water-Uptake (W.U.), λ, and Crosslink Rates (D_crosslink_)

The ion exchange capacity (IEC) was defined as milliequivalents of sulfonic groups per gram of dried sample. A piece of membrane was soaked in 20 mL of a 2 M NaCl solution and equilibrated for more than 24 h to replace the protons with sodium ions. The solution was then titrated with a 0.01 M NaOH solution. The IEC was calculated using the following equation: IEC (meq/g) = c*v*/W_dry_, where c (mmol/L) is the concentration of the standardized NaOH aqueous solution used for titration (0.01mol/L), *v* (L) is the volume of the standardized NaOH aqueous solution used for titration, and W_dry_ (g) is the mass of the dry membrane. The degree of sulfonation (D.S.) of the membranes was calculated using the following equation: D.S. (Sulfonic acid group/repeating unit; R.U.) = [IEC/1000 × F_w_ (R.U.)]/[1 − (IEC/1000 × F_w_ (SO_3_))], where F_w_ (R.U.) = 400.45 and F_w_ (SO_3_) = 80.06.

The water-uptake (W.U.) of the membranes at room temperature was calculated using the following: W.U. (%) = [(W_wet_ − W_dry_)/W_dry_] × 100, where W_wet_ is the mass of the wet membrane. The membranes were cut into 10 mm × 10 mm squares and dried for >24 h at 80 °C in a dry oven. The membranes were immersed in boiling DI H_2_O for >1 h before the measurements.

The hydration number (λ) for the membranes was calculated using the following: λ ([H_2_O]/[SO_3_H]) = [1000(W_wet_ − W_dry_)]/18 W_dry_IEC.

The degree of crosslinking (crosslink rate, D_crosslink_) in the membranes was calculated using the following: D_crosslink_ (%) = [(IEC_before annealing_ − IEC_after annealing_)/IEC_before annealing_] × 100.

### 2.6. Oxidative Stability (Fenton’s Test)

The oxidative stabilities of the membranes were evaluated by immersing a small piece of sample into Fenton’s reagent [3 wt% H_2_O_2_ and 2 ppm Fe(II) (added as FeCl_2_·4H_2_O)] at 80 °C for 1 h while stirring. The samples were dried at 60 °C in a vacuum oven before the measurements. The membranes were washed with DI H_2_O repeatedly and dried in a vacuum oven at 60 °C overnight after the reaction. The oxidative stabilities were determined as follows: 100[(mass of residual membrane after the test)/(initial mass of membrane)].

### 2.7. Thermal Behavior

The thermal and mass properties of the membranes were investigated using thermogravimetric and mass analysis with a Discovery TGA/MS (TA instruments). The samples were heated from room temperature (RT) to 600 °C at 20 °C/min in an He atmosphere.

The dynamic elastic modulus and dissipation (tan δ) of the membranes were measured by DMA Q800 (TA Instruments) at 1 Hz in the temperature range of 20–200 °C at a heating rate of 5 °C/min.

### 2.8. Mechanical Behavior

The stress–strain tests on the membranes were performed using a Tension Test Machine (Shimazu, EZ-S) at room temperature with a constant crosshead speed of 5 mm/min. The samples were cut using the super dumbbell cutter SDMP-1000 (Dumbbell Co., Japan). The membrane thicknesses were measured using a Mitutoyo 547-401 ABSOLUTE Digimatic Thickness Gauge, Japan.

### 2.9. Membrane Nanostructure

Small angle X-ray scattering (SAXS) measurements were performed in the beamline BL15A2 of the Photon Factory in KEK (Tsukuba, Japan). The X-ray beam was monochromatized to 1.2 Å with a curved monochromator (triangular Ge (111) crystal with the asymmetric angle of 8.0 degrees). The membrane samples were put on the stage of a rotational autochanger, and the measurements were performed on the membranes under wet conditions, for which a small amount of water was added dropwise. Scattered photons were detected with a two-dimensional semiconductor detector (PILATUS3 1M, 981 × 1043 pixels, pixel size: 0.172 mm × 0.172 mm, Dectris, Switzerland) with a camera distance of 170 cm, and the signals were accumulated for 30 s. The image was processed by using the SAngler program. The image was calibrated with the diffraction image of silver behenate and then averaged over azimuthal angle to obtain a one-dimensional profile with the subtraction of SAXS profile of air.

### 2.10. Conductivity Measurements

The proton conductivities of the membranes were determined using four-point probe impedance spectroscopy. Through-plane proton conductivities of the membranes using the MTS cell head were measured as a function of the relative humidity (RH, 20–90%) in the temperature range of 40–120 °C, using a 740 membrane test system (MTS, Scribner Associates, Inc.) with a phase sensitive multimeter (model PSM1735, Newtons4th Ltd.) combined with an impedance analysis interface. A frequency range of 1 Hz–1 MHz and a peak-to-peak voltage of 10 mV were used during the impedance measurements. The samples were equilibrated in a temperature and humidity chamber at specified temperatures and relative humidities (RHs) for 30 min before the measurements. The chamber was pressurized at 130 kPa for the measurements in the temperature range of 100–120 °C.

### 2.11. Membrane, Gas Diffusion Layer, and Catalyst Electrode Information for MEA

The thickness of the membranes was about 50 µm. A Pt/C/ionomer (ionomer/carbon = 1) catalyst electrode containing 0.3 mg/cm^2^ of Pt on a GDL electrode (Sigaracet^®^ GDL 25BC of SGL Group Co. Ltd., Japan) was purchased from EIWA Corporation.

### 2.12. Preparation of Membrane Electrode Assembly (MEA)

The effective electrode area of the single cell was 4 cm^2^. A hot press machine (MODEL A-010D, FC-R&D Company, Sagamihara, Japan) was used. The MEA was prepared on Cu plates. Polyimide films were used between the Cu plates to facilitate peeling of the MEA. The MEA was obtained by inserting a membrane between the anode and cathode and hot-pressing at 130 °C at ~9.8 kN for 20 min.

### 2.13. Single Fuel-Cell Performance and Durability

A single cell (active area: 20 × 20 mm^2^) was purchased from Ulimeng Eng Co., Ltd., Korea. The fuel cell system was evaluated using AutoPEM of Toyo Corporation, Japan. The current-voltage (I-V) performance was measured in relation to the amount of hydrogen and oxygen at the anode and cathode, respectively, at 80 °C, 100% and 60% RH, and ambient pressure. The gas utilizations at the anode and cathode were 56% and 14%, respectively. The I-V characteristics of a single cell were obtained by activating at 1 A for 20 h, increasing the current from 0.02 to 6 A, while maintaining a constant current value for 5 min, and sweeping 5 times. Next, the cell was purged for 10 h with nitrogen gas, and cyclic voltammetry (CV) and linear sweep voltammetry (LSV) were performed in the potential range of 0.02–0.9 V at a scan rate of 50 mV/s and 0.02–0.5 V at 2 mV/s, respectively. During the measurements, 100 mL/min humidified nitrogen and 100 mL/min hydrogen were fed to the working electrode and counter electrode, respectively. The effective catalytic area (ECA) from the results (oxidation of adsorbed H) of the CV was calculated using the following equation: ECA ([cm^2^/g] = (charge density, µC/cm^2^)/[210 (µC/cm^2^) × electrode loading, (g/cm^2^)]. A long-term test using a constant current method was performed at 80 °C and 100% RH using a PEMTest8900 from Toyo Corporation.

## 3. Results and Discussion

### 3.1. Activation of the CSPPSU Membranes

Figure 1 shows an image (16 × 18 cm^2^) (Figure 1a) and the chemical structure (Figure 1b) of the chemically crosslinked SPPSU (CSPPSU) membranes. CSPPSU membranes were prepared by using a heat treatment method as previously reported [34,36,37]. Crosslinking between the sulfo groups (–SO_3_H) by heat treatment can produce undesirable byproducts, such as sulfur dioxide (SO_2_), which do not completely evaporate from the membrane. The remaining SO_2_ was found to decrease the original membrane characteristics obtained from the MEA process, causing bad fuel cell performances. We removed SO_2_ by activating the membranes with an alkaline solution. In addition, it was found that the mechanical and chemical properties and the durabilities of the membranes were dramatically stabilized after our activation process.

Figure 2 shows the thermal stabilities of synthesized SPPSU, CSPPSU after heat treatment at 180 °C, and CSPPSU membranes activated without and with an NaOH solution. The thermal stabilities of the CSPPSU membranes after heat treatment at 180 °C were better than those of SPPSU. There was no significant difference in the thermal stabilities of the CSPPSU membranes after the heat treatment at 180 °C and the CSPPSU membrane activated without NaOH solution. On the other hand, the stability of the CSPPSU membranes was much higher after activation treatment with NaOH. Mass analysis indicated that the stability of the CSPPSU membrane was due to the removal of the SO_2_ trapped in the membrane. From these results, it is suggested that activation with an alkaline solution is necessary when crosslinked membranes are obtained using DMSO as a solvent.

### 3.2. Mechanical Properties of the CSPPSU Membranes

Figure 3 shows the dynamic elastic modulus and Tan δ properties of CSPPSU and Nafion212 membranes with respect to temperature. The dynamic elastic modulus of CSPPSU membranes was constant up to 200 °C, whereas that of the Nafion212 membrane decreased due to changes in the molecular structure with an increase in the temperature. From Tan δ, the glass transition temperature of the Nafion212 membrane was around 100 °C, and that of the CSPPSU membrane was >200 °C. This result suggests that the CSPPSU membrane is suitable for application as a high-temperature electrolyte membrane over 100 °C.

Figure 4 shows the tensile strengths and elongation results for the CSPPSU and Nafion212 membranes at room temperature. The Nafion212 membrane had a low tensile strength but very high tensile elongation. The tensile strength of the CSPPSU membranes was higher than that of the Nafion212, but the tensile elongation was lower than that of the Nafion212. On the other hand, the tensile strength and elongation of the CSPPSU membranes were similar or better than those of other hydrocarbon-based polymer electrolyte membranes [26,28,34,37,50]. However, an increase in the tensile elongation (high flexibility) is required to obtain good contact, which can reduce the contact resistance and proton transfer in the membrane at the interface between the membrane and the electrode. The mechanical and thermal properties of PPSU, CSPPSU and Nafion212 membranes are summarized in Table 1.

### 3.3. Nanostructures of the CSPPSU Membranes

The nanostructures of wet CSPPSU membranes were characterized by using small angle X-ray scattering (SAXS) (Figure 5) [4]. An intensity maximum was observed at q = 1.6 nm^−1^ (q = 4πsinθ/λ). The average spacing, L, of the scattering objects was calculated from the peak position by using the Bragg equation (L = 2π/θ) to be 3.9 nm. This value was lower than the L value for Nafion (5 nm) [4,51,52]. In addition, the peak intensity of the CSPPSU membrane was lower than that of Nafion. In general, in a Nafion membrane, there is good separation between hydrophilic and hydrophobic phases and it has a good ion conduction path in a humidified state. On the other hand, it is known that hydrocarbon-based membranes, such as SPEEK, have a longer distance between sulfone groups and less continuity of the conduction paths than Nafion membranes do [4]. From these results, we believe that the CSPPSU membranes have good conduction paths and that the IEC is high (Table 2). However, the volume density of the ion clusters in the nanostructures is low, and the SAXS peak intensity is low.

### 3.4. Proton Conductivities of the CSPPSU Membranes

Since SPPSU was crosslinked by using heat treatment, it was difficult to precisely control the crosslinking in the nanostructure. However, using this method, stable and macroscopically reproducible CSPPSU membranes were obtained. Figure 6 shows the proton conductivities of CSPPSU membranes obtained from the average value of the conductivities of four different membranes in relation to the temperature and relative humidity. These conductivities were lower than those obtained in our previous studies [34,36,37] due to different synthetic and activation conditions. The conductivities of the CSPPSU membranes were lower than those of the Nafion212 membranes [5,37,52] and were lower by one order of magnitude under low relative humidification conditions (Table 2). The reasons for the difference include the following: the ion cluster size of the CSPPSU membrane is small, the distance between the sulfone groups due to the backbone structure of PPSU is long, and the number of continuous conduction paths is small, as described previously. The design of CSPPSU membranes with high conductivities and high mechanical stabilities is required.

### 3.5. IEC, W.U., λ, and D_crosslink_ Values of CSPPSU Membranes

In Table 2, the characteristics of the CSPPSU membranes are summarized in comparison with Nafion212 membranes. The IEC of the CSPPSU membranes was higher than that of Nafion212, but W.U. and λ were lower than those of Nafion212. This is significant for hydrocarbon-based electrolyte membranes because high IECs tend to cause swelling of the membrane in water and thus deteriorate the stability of the membrane. The CSPPSU membranes had a crosslink rate (D_crosslink_) of 47.3% and high mechanical stabilities with a high IEC and reasonable W.U. values. These results indicate that the crosslinking method is effective for increasing the mechanical and chemical stabilities of hydrocarbon electrolyte membranes.

### 3.6. Oxidative Stability of the CSPPSU Membranes

The degradation phenomena of the electrolyte membrane in a fuel cell are thought to be caused by the following [47]. One is degradation on the anode side due to H radicals and O_2_ diffused from the cathode side through the electrolyte membrane. The other is degradation on the cathode side, where HO or HO_2_ radicals from O_2_ reduction diffused on the cathode side. However, the HO and HO_2_ radicals on the cathode side are the main causes of electrolyte membrane degradation. The oxidative stabilities of the CSPPSU membranes were evaluated by using an ex situ method wherein they were treated with Fenton’s reagent (3 wt% H_2_O_2_ + 2 ppm Fe (II)) at 80 °C for 1 h. The CSPPSU membranes had stabilities in the range of 91.3–99.4% (Table 2). However, the membranes broke when rubbed by hand after the Fenton tests. A method for improving the oxidative stabilities while improving the proton conductivities of the CSPPSU membrane is currently under study and will be reported later.

### 3.7. Fuel Cell Properties Using CSPPSU Membranes

Evaluation of a fuel cell is complicated because the membrane, catalyst electrodes containing an ionomer, the interfaces between a membrane and an electrode and between the electrode and the separator, and gas supply affect the power generation reaction. However, the current-voltage (I-V), cyclic voltammetry (CV), and hydrogen gas crossover characteristics of a single cell were evaluated under similar conditions but with different electrolyte membranes. Figure 7 and Figure 8 show the I-V_iR free_ and CV characteristics of the CSPPSU membranes and the Nafion212 membrane evaluated at a cell temperature of 80 °C and 100% and 60% RH, respectively. The voltage differences using the CSPPSU and Nafion212 membranes were 37 mV at 1 A/cm^2^ and 100% RH and 33 mV at 0.5 A/cm^2^ and 60% RH. On the other hand, the cell resistances using the CSPPSU and Nafion212 membranes were 73 mohm and 24 mohm at 100% RH and 161 mohm and 42 mohm at 60% RH, respectively (Table 3). The I-V_iR free_ results show that the difference in the conductivities of the CSPPSU and Nafion212 membranes was large (Table 2), whereas the difference in the fuel cell was not so large. However, on the basis of the resistance value of the entire cell, the difference between the CSPPSU and the Nafion212 membranes was large, especially when RH was 60%. This means that the interface resistance between the CSPPSU membrane and the catalyst electrodes was large and that the resistance due to phase separation between the hydrocarbon-based CSPPSU membrane and the fluorine-based Nafion ionomer was large. Moreover, the CV characteristics show that the catalytic activity on the anode side and the cathode side of the cell containing the CSPPSU membrane was lower than that of the cell containing the Nafion212 membrane (Table 3). It is possible that the catalyst has been poisoned by the CSPPSU membrane.

### 3.8. H_2_ Crossover of the CSPPSU Membranes

Figure 9 shows hydrogen gas crossover characteristics for MEA using the CSPPSU and Nafion212 membranes obtained using LSV [53]. The current densities of the MEA using the CSPPSU and the Nafion212 membranes at 0.4 V were 0.08 and 1.24 mA/cm^2^, respectively (Table 3). From the literature [53], gas crossover occurs when the current density is >2 mA/cm^2^, and short circuit occurs when the current density is >5.5 mA/cm^2^. The current density of the Nafion212 membrane was in good agreement with literature values [53]. On the other hand, the current density of the CSPPSU membranes is 15 times smaller than that of the Nafion212 membrane. The current density of the CSPPSU membranes was lower than those using other hydrocarbon membranes [40], indicating that the CSPPSU membrane has high durability against gas crossover.

### 3.9. Durability Using CSPPSU Membranes

Figure 10 shows a plot of the voltage vs. time for MEAs using CSPPSU and Nafion212 membranes with an applied current of 1 A at a cell temperature of 80 °C and 100% RH. The MEA cell using the Nafion212 membrane was very stable during the measurement. On the other hand, the MEA cell using the CSPPSU membrane was a little unstable in the initial stage but became stable by 4000 h. However, the voltage gradually decreased after 4000 h. The single cell was separated, and the MEA was inspected. There were no cracks or pinholes in the membrane where no electrode was attached, and the appearance was the same as that before the test. However, the portion between the membrane and the electrode could not be observed because the membrane and the electrode were firmly attached to each other. However, there is a possibility that the voltage may have decreased due to mechanical deterioration, such as a crack or pinhole, between the membrane and the electrode. Moreover, it is conceivable that HO or HO_2_ radicals formed during the reduction of oxygen on the cathode side may chemically degrade the CSPPSU membrane. There are few reports on the long-term stabilities of fuel cells using hydrocarbon-based electrolyte membranes, and we believe this is the first time that the long-term stability of a cell using a CSPPSU membrane has been evaluated for 4000 h. We are currently conducting more detailed evaluations of MEA cells and developing CSPPSU membranes with long-term stabilities like Nafion membranes.

## 4. Conclusions

High-IEC SPPSU was synthesized using commercially available PPSU, and a large-area crosslinked membrane was produced. In addition, we developed an activation process using alkaline and acidic solutions to remove the remaining sulfur dioxide (SO_2_) formed as a byproduct during the heat treatment. CSPPSU membranes activated with our method were compared with a Nafion212 membrane. The proton conductivities of the CSPPSU membranes were lower than that of the Nafion212 membrane, but the glass transition temperature and tensile strength were higher. This indicated that the CSPPSU membrane could be used even at high temperatures. From SAXS analysis, the CSPPSU membrane had an ion cluster structure that could act as a proton conduction path. The I-V_iR free_ characteristics of MEA using the CSPPSU membrane were very good and similar to those when a Nafion212 membrane was used. The durability towards hydrogen gas crossover using the CSPPSU membranes was better than that using the Nafion212 membrane. In the long-term evaluation by using a constant current method, a stability of 4000 h was obtained for the first time. On the other hand, from CV, the resistance between the membrane and the electrode and the poisoning of the catalyst were larger than those when the Nafion212 membrane was used. Thus, further analyses and higher performances of MEAs are required.

## Figures and Tables

**Figure 1 membranes-10-00031-f001:**
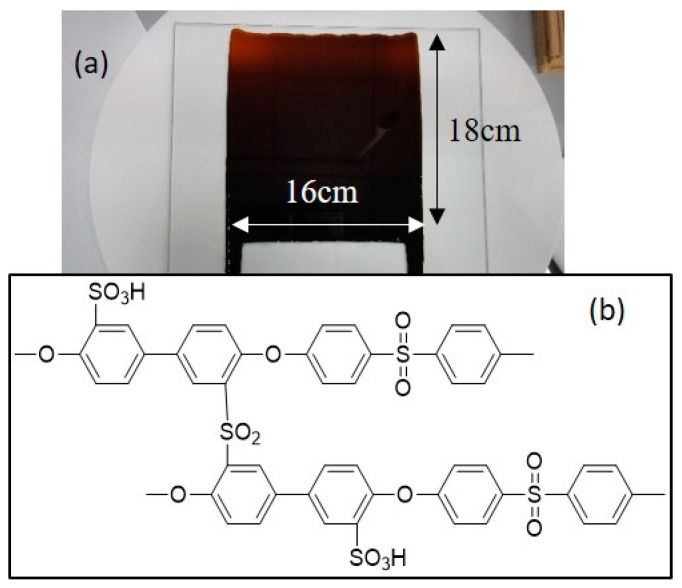
(**a**) Image of a crosslinked sulfonated polyphenylsulfone (CSPPSU) membrane on a glass plate and (**b**) chemical structure of a CSPPSU.

**Figure 2 membranes-10-00031-f002:**
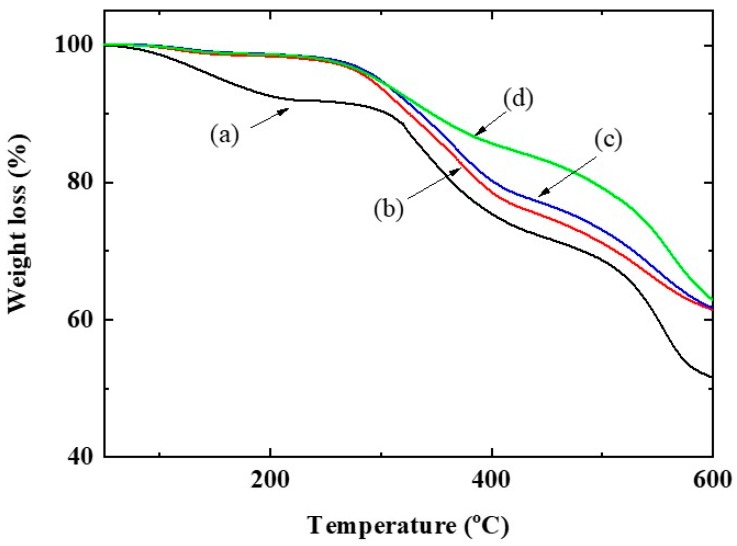
Thermal stabilities in a helium gas flow at a heating rate of 20 °C/min of (**a**) synthesized sulfonated polyphenylsulfone (SPPSU), (**b**) CSPPSU after the heat treatment at 180 °C, and CSPPSU membranes activated (**c**) without and (**d**) with NaOH solution process.

**Figure 3 membranes-10-00031-f003:**
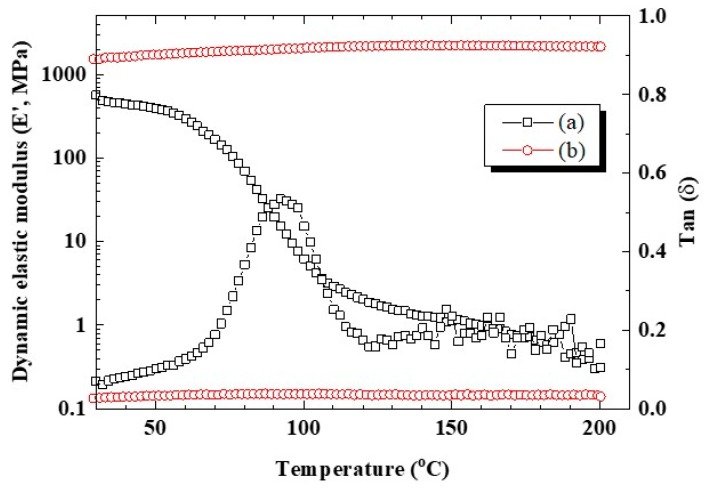
Dynamic elastic modulus and Tan δ of (**a**) Nafion212 and (**b**) CSPPSU membranes at 1 Hz.

**Figure 4 membranes-10-00031-f004:**
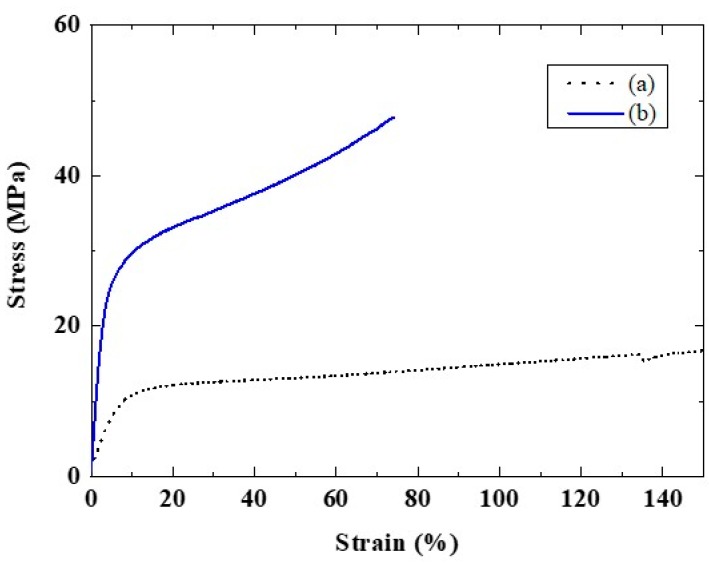
Stress–strain properties of (**a**) Nafion212 (t = 0.050 mm) and (**b**) CSPPSU (t = 0.066 mm) membranes at room temperature.

**Figure 5 membranes-10-00031-f005:**
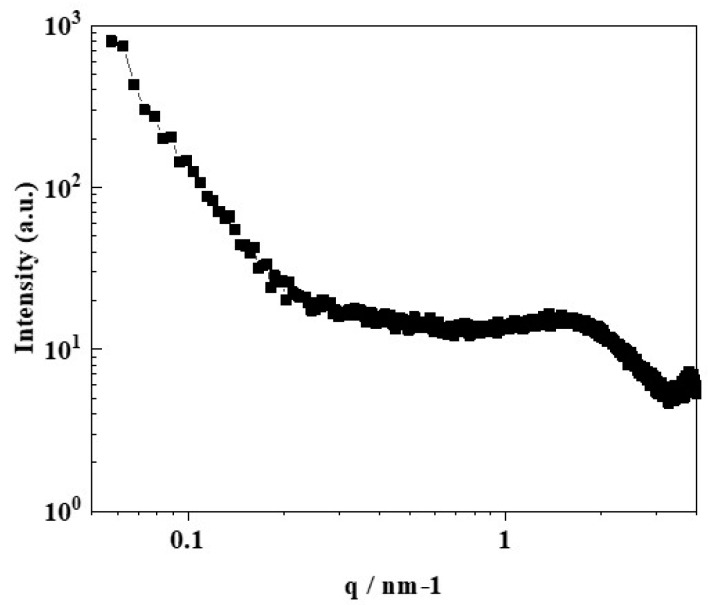
Small angle X-ray scattering (SAXS) profile of a wet-CSPPSU membrane at room temperature.

**Figure 6 membranes-10-00031-f006:**
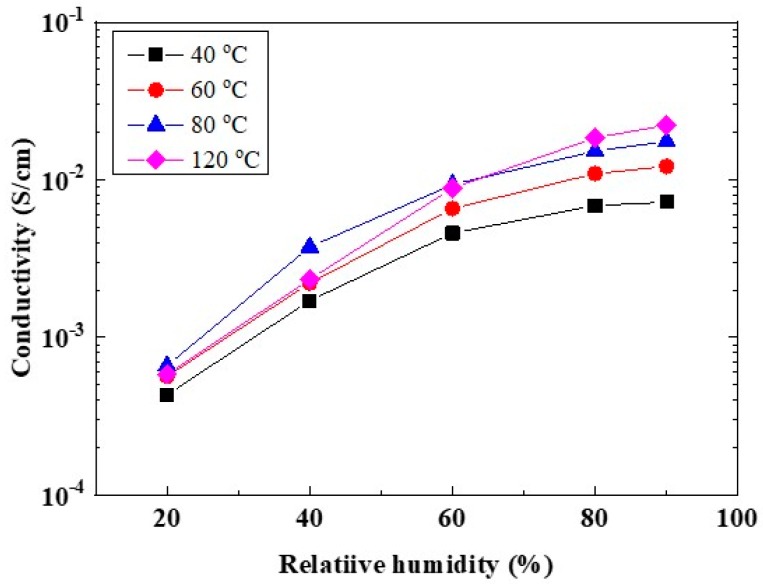
Proton conductivities of the CSPPSU membranes vs. the relative humidity at the specified temperatures.

**Figure 7 membranes-10-00031-f007:**
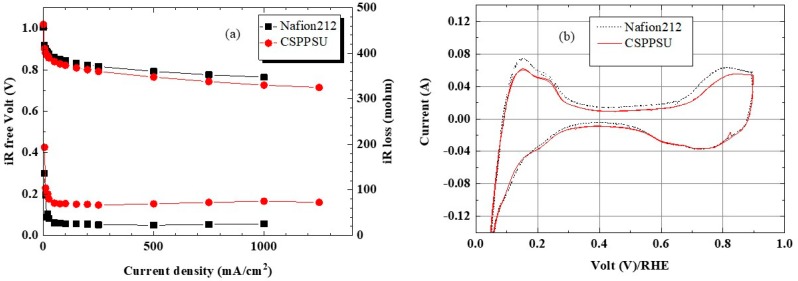
(**a**) I-V_iR free_ and (**b**) CV properties of CSPPSU and Nafion212 membranes at 80 °C and 100% RH.

**Figure 8 membranes-10-00031-f008:**
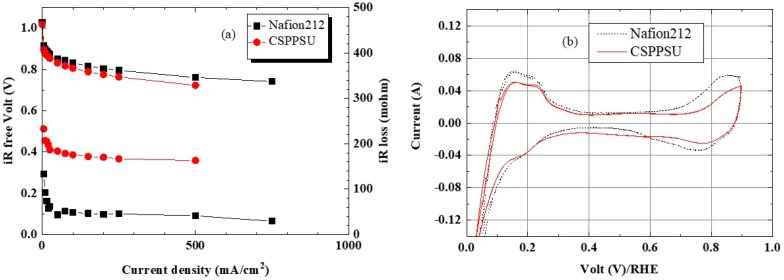
(**a**) I-V_iR free_ and (**b**) CV properties of CSPPSU and Nafion212 membranes at 80 °C and 60% RH.

**Figure 9 membranes-10-00031-f009:**
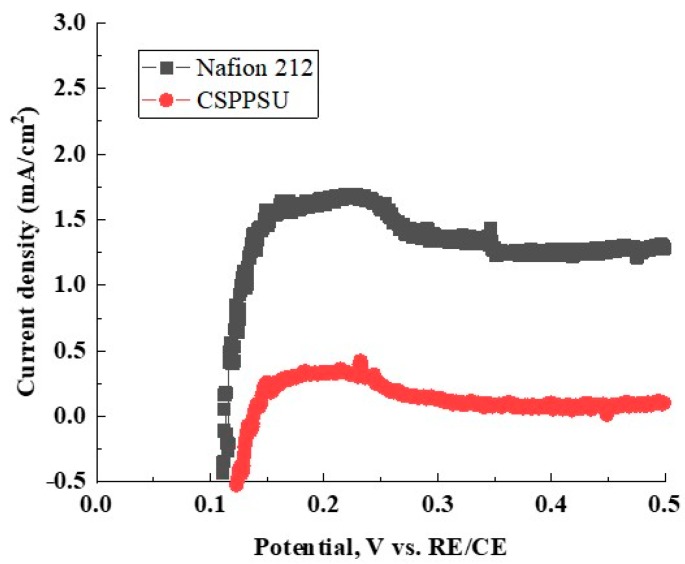
Hydrogen crossover properties of CSPPSU and Nafion212 membranes at 80 °C and 100% RH.

**Figure 10 membranes-10-00031-f010:**
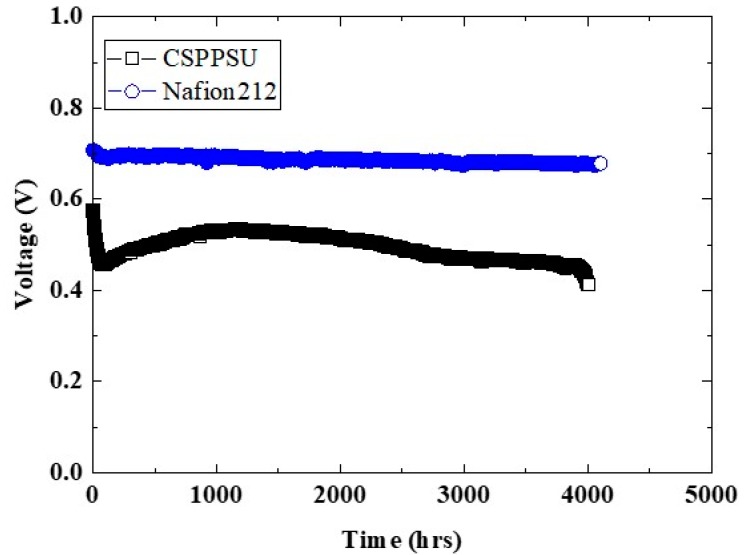
Durability of CSPPSU and Nafion212 membranes at 80 °C, 100% RH, and 1 A.

**Table 1 membranes-10-00031-t001:** Mechanical and thermal properties of PPSU, CSPPSU and Nafion212 membranes.

Polymer Membrane	PPSU (Solvay) *	CSPPSU	Nafion212
Tensile modulus (MPa)	2340	2000	~500
Tensile strength (MPa)	69.6	48	16 at 150%
Tensile elongation (%) (break)	60–120	74	>150
Flexural modulus (MPa)	2410	757	155
Glass transition (°C)	220	>200	~100
Decomposition (°C)	540	~300	~300

* From Solvay PPSU R-5000_Datasheet.

**Table 2 membranes-10-00031-t002:** Physicochemical and conductivity properties of CSPPSU and Nafion212 membranes.

Membrane			CSPPSU	Nafion212
IEC (meq/g)			~2	0.9
W.U. (%)			43	50
λ			12	31
D_crosslink_ (%)			47.3	-
R_oxidation_ (%)			91.3–99.4	100
Cluster size (nm)			3.9	5.0
Conductivity (mS/cm)	80 °C	20% RH	0.7	7.3
		90% RH	18	80
	120 °C	20% RH	0.6	6.6
		90% RH	22	105

**Table 3 membranes-10-00031-t003:** I-V and H_2_ crossover data for single cells using CSPPSU and Nafion212 membranes.

Membrane	80 °C, 100% RH			80 °C, 60% RH	
	iR Loss (mohm) at 1 A/cm^2^	ECA (m^2^/g)	H_2_ Crossover (mA/cm^2^) at 0.4 V	iR Loss (mohm) at 1 A/cm^2^	ECA (m^2^/g)
CSPPSU	73	82	0.085	161	73
Nafion212	24	99	1.24	42	92

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
