# Peer review of "Chemically Crosslinked Sulfonated Polyphenylsulfone (CSPPSU) Membranes for PEM Fuel Cells"

_membranes, 2020, doi:10.3390/membranes10020031_

Round 1

Reviewer 1 Report

The authors show an original method and characterizes their CSPPSU membrane. While the results is relevant for both PEM fuel cells and electrolyzers, the presentation of the results can be improved. The paper is well written and should arouse the interest of readers with an electrochemical background. However, minor corrections should be done before acception.

Author Response

Answers to Reviewer 1

I would like to thank for your kind reading of our paper. According to the comments, I have revised our manuscript with yellow highlighter.

Reviewer 2 Report

The article presents a novel type of treatment to clean crosslinked membranes of SPPSU and present a complete series of characterizations to deeply understand the effect of this treatment on the performances of the membrane. The article is well written and follow a good scientific investigation, presenting a lot of different characterizations (ex situ and in situ) of the membrane to demonstrate the effectiveness of the treatment. A good comparison with Nafion is done.
Some corrections were done (see the text attached) to improve and valorise the content of the article.
Some questions are also reported to the authors. However, I report here some questions that are significant to me:
1. From the introduction to fuel cell tests, have the authors check if the SO2 moieties can be a cause of the deactivation of Pt? Can be this SO2 responsible of the low fuel cell test performances with CSPPSU?
2. To compare CSPPSU and Nafion it is good to make the same treatment, why you perform two different treatments?
3. Concerning the water uptake, maybe the treatment at 80 °C it is not enough because some water can remain in the membrane, did you made some treatment at higher temperature (100 °C) or in a desiccator to compare?
4. Concerning the TGA analysis, why you did the test in He atmosphere and not in air to simulate the real degradation? In the future for others tests you can reduce also the speed to have more precision. Please show the mass analysis, can be interesting to identifies the species of each mass loss.
5. Do you repeat the IEC evaluation before and after the treatment to check if you have no degradation?
6. Concerning the stress/strain test please specify if it is before or after the treatment.
7. Concerning the conductivity, please specify the IEC and degree of crosslink of the showed membrane. Can you also report the errors on your 4 measures?

Others little questions are reported in the text.
Thank you for this articles that deeply enter in the comprehension of the crosslinking and in the possible products that can affect the performances of the aromatic based membranes.

Author Response

Answers to Reviewer 1

I would like to thank for your kind reading of our paper. According to the comments, I have revised our manuscript with yellow highlighter. Answers to the reviewer are listed in the attached PDF file.
